# Remote sensing of plant trait responses to field-based plant-soil feedback using UAV-based optical sensors

Bob van der Meij[1], Lammert Kooistra[2], Juha Suomalainen[2], Janna M. Barel[3], Gerlinde B. De Deyn[3]

[1]Faculty of Geosciences, Utrecht University, P.O. Box 80.115, 3508 TC Utrecht, the Netherlands
[2]Laboratory of Geo-Information Science and Remote Sensing, Wageningen University and Research, P.O Box 47, 6700 AA Wageningen, the Netherlands
[3] Department of Soil Quality, Wageningen University and Research, P.O Box 47, 6700 AA Wageningen, the Netherlands *Correspondence to*: Bob van der Meij (bobvandermeij@gmail.com)[1] or Gerlinde B. De Deyn (gerlinde.dedeyn@wur.nl)[3]

**Abstract.** Plant responses to biotic and abiotic legacies left in soil by preceding plants is known as plant-soil feedback (PSF). PSF is an important mechanism to explain plant community dynamics and plant performance in natural and agricultural systems. However, most PSF studies are short-term and small-scale due to practical constraints for field scale quantification of PSF effects, yet field experiments are warranted to asses actual PSF effects under less controlled conditions. Here we used Unmanned Aerial Vehicle (UAV)-based optical sensors to test whether PSF effects on plant traits can be quantified remotely. We established a randomized agro-ecological field experiment in which six different cover crop species and species combinations from three different plant families (*Poaceae*, *Fabaceae*, *Brassicaceae*) were grown. The feedback effects on plant traits were tested in oat (*Avena sativa*) by quantifying the cover crop legacy effects on key plant traits: height, fresh biomass, nitrogen content and leaf chlorophyll content. Prior to destructive sampling, hyperspectral data was acquired and used for calibration and independent validation of regression models to retrieve plant traits from optical data. Subsequently, for each trait the model with highest precision and accuracy was selected. We used the hyperspectral analyses to predict the directly measured plant height (RMSE= 5.12 cm, $R^2$= 0.79), chlorophyll content (RMSE= 0.11 g m$^{-2}$, $R^2$= 0.80), N-content (RMSE= 1.94 g m$^{-2}$, $R^2$= 0.68), and fresh biomass (RMSE= 0.72 kg m$^{-2}$, $R^2$=0.56). Overall the PSF effects of the different cover crop treatments based on the remote sensing data matched the results based on *in situ* measurements. The average oat canopy was tallest and its leaf chlorophyll content highest in response to legacy of *Vicia sativa* monocultures (100 cm, 0.95 g m$^{-2}$, respectively) and in mixture with *Raphanus sativus* (100 cm, 1.09 g m$^{-2}$, respectively), while the lowest values (76 cm, 0.41 g m$^{-2}$, respectively) were found in response to legacy of *Lolium perenne* monoculture, and intermediate responses to the legacy of the other treatments. We show that PSF effects in the field occur and alter several important plant traits that can be sensed remotely and quantified in a non-destructive way using UAV-based optical sensors; these can be repeated over the growing season to increase temporal resolution. Remote sensing thereby offers great potential for studying PSF effects at field scale and relevant spatial-temporal resolutions which will facilitate the elucidation of the underlying mechanisms.

**Keywords**: plant-soil feedback, soil legacy, treatment discrimination, high-resolution hyperspectral imagery, UAV remote sensing, plant height, biomass, nitrogen, leaf chlorophyll

# 1 Introduction

## 1.1 Plant-soil feedback

Plants influence biotic and abiotic soil properties and these changes can last in soil even after the plant is no longer there. This soil legacy of plants can feedback to the performance of subsequently grown plants such that their growth is enhanced or suppressed relative to growth in soil without a plant legacy (Brinkman et al., 2010). In recent years there has been a growing interest for understanding this mutual interaction process known as plant-soil feedback (PSF), because of its importance as a mechanism to understand plant community dynamics such as plant succession, exotic plant invasion and biodiversity-productivity relations (Kulmatiski et al., 2008; Bever et al., 2010; van der Putten et al., 2013). Also in agricultural systems, PSF is highly relevant and is one of the main reasons for practicing crop rotations so that the risk for negative PSF can be kept low, however mechanistic understanding of PSF is needed in order to make use of the potential of generating positive PSF effects (van der Putten et al., 2013; Dias et al., 2014). The vast majority of PSF studies has been conducted under highly controlled laboratory or greenhouse conditions and in order to adequately assess the impact of PSF in real ecosystems there is an urgent need to test PSF in the field and develop methodologies that facilitate this (Kulmatiski and Kardol 2008; van der Putten et al., 2013). At the same time agronomic field studies are still vastly reliant on labor-intensive, time-consuming, destructive, and selective *in situ* data collection by experts (Nebiker et al., 2008). It is argued here that advancements in remote sensing platforms (i.e. Unmanned Aerial Vehicles) and imaging spectroscopy combined may offer novel opportunities for non-invasive assessment of plant trait responses to soil legacies at enhanced spatial-temporal resolutions (Faye et al., 2016) and unprecedented detail (Fiorani and Schurr, 2013), with potential use in both fundamental and applied research in natural and agro-ecosystems.

## 1.2 Plant traits and remote sensing

During the last decades, plant trait based ecology has developed fast and is enabling a better mechanistic understanding of ecosystem processes across spatial and temporal scales (Cornelissen et al., 2003; Wright et al., 2004; Kattge et al., 2011; Diaz et al., 2016). Important plant traits from the perspective of ecosystem functioning comprise of physical and chemical plant characteristics such as plant stature and plant N content (Cornelissen et al., 2003; Diaz et al., 2016). Trait based studies in plant ecology have mostly focused on natural ecosystems and their responses to natural and human-imposed disturbances (e.g., Garnier et al., 2007; de Bello et al., 2010). However, recently trait based approaches are being used to predict plant legacy effects in soil and subsequent plant responses (Orwin et al., 2010; Ke et al., 2015; Cortois et al., 2016). To date most PSF experiments have focused on plant biomass as sole measure of plant responses to soil legacies. However, there are a number of plant traits that are highly relevant for plant performance in both agricultural and natural systems as they represent aspects of plant quality (N content), competitive ability (plant height) and potential activity (chlorophyll content) which are also highly relevant for plant growth modeling.

Plant attributes invoke diverging interactions (i.e. absorption, reflection and transmission) with light over different wavelengths (Pinter et al., 2003; Homolová et al., 2013). Consequently, spectral remote sensing has proven an effective source of information for monitoring vegetation in the field, non-invasively and comparatively efficiently, for diversified applications in past agronomic and ecological studies (Jones &

Vaughan, 2010; Thenkabail et al., 2012), including species classification (Franklin, 2001), quantification of biophysical or biochemical plant constituents (Mulla, 2013; Qi et al, 2012), and multi-temporal monitoring of plant development (Zhang et al., 2003). Advancements in imaging spectroscopy are particularly relevant in this respect (Ortenberg, 2012; Fiorani and Schurr, 201). Imaging spectrometers allow detection of subtle variations in spectral reflectance of the plant canopy by acquiring data in large numbers (up to hundreds) of contiguous narrow spectral bands (Campbell and Wynne, 2002; Warner et al., 2009; Qi et al., 2012;). They invoke increased sensitivity to multiple crop traits (Homolová et al., 2013) and are therefore superior to multispectral alternatives (Shippert, 2004; Govender et al., 2007) regarding accurate discriminatory mapping and retrieval of vegetation traits (Rascher et al., 2011; Thenkabail et al., 2012; Kooistra et al., 2014). A variety of vegetation indices (VIs), embodying a mathematical manipulation of raw spectra from two or more wavelengths, have been conceived for vegetation monitoring purposes (Goswami et al., 2015) and were demonstrated to be stronger related to distinct plant traits than individual wavelengths due to isolation and enhancement of the spectral signal (Chuvieco, 2011).

### 1.3 Remote sensing with UAV

It has been argued that conventional ground-based, airborne, or space-borne platforms are largely unable to provide remote sensing data at an adequate spatial (cm-level) and/or spectral resolution, repeatedly and at affordable costs for small-scale crop and vegetation field experiments with a large number of individual plots (Berni et al., 2009; Zhang & Kovacs, 2012; Colomina and Molinda, 2014). Unmanned Aerial Vehicles (UAVs), providing access to images with sufficiently high and flexible spatial-temporal resolutions at competitive costs and at an acceptable operational resilience, have received increased attention in related fields such as agriculture (Berni et al., 2009; Rango et al., 2009; Zhang and Kovacs, 2012; Honkavaara et al, 2013), and plant phenotyping (Chapman et al., 2014; Haghighattalab et al., 2016). Furthermore, proper plant trait retrieval methods and the associated accuracy (i.e. geometric and/or radiometric) and resolution(s) thereof require thorough evaluation (Lelong et al., 2008; Hardin and Jensen, 2011; Hruska et al., 2012).

### 1.4 Objectives and hypotheses

The objective of the present study was to (i) develop and demonstrate a methodology for plant trait analyses using UAV based imaging spectroscopy data, (ii) to assess the resultant accuracy for plant trait retrieval, and (iii) to evaluate the ability to discriminate plant trait responses to different plant legacies in soil in a field-based PSF experiment using UAV based imaging spectroscopy data. We expected that UAV-based optical sensors can detect and quantify the plant traits (height, fresh biomass, N content, C content and leaf chlorophyll content) at adequate resolution and accuracy. We also expected that plant trait responses to plant legacies quantified via *in situ* (i.e. ground-based) measurements can be assessed as well using UAV imaging spectroscopy analyses.

### 2 Materials and methods

The investigation was conducted within a large-scale field experiment (Barel et al. in prep.), aimed at uncovering the influence of legacies of various major crop species and combinations of cover crops on succeeding plants. A UAV campaign and corresponding destructive sampling were conducted to retrieve airborne imaging spectroscopy data and *in situ* oat (*Avena sativa*) plant trait data, respectively. In this study we focused on effects

and characterization of traits for oat as it has an erect growth form and a relatively long growing season which increased the likelihood of being responsive to soil legacies. Moreover, oat is related to many important grain crops as well as to grassland species in managed and natural systems, hence it has the potential to serve as a model for future experiments. Spectral measurements were fitted to plant trait data to calibrate relationships between both variables, which were subsequently assessed with respect to prediction and discrimination accuracy through independent validation.

## 2.1 Study area

The field experiment was established in spring 2014 (Barel et al., in prep) to investigate the legacy of various species and species combinations of cover crops on subsequently grown main crops. The study site (Fig. 1) is located at the agricultural field facilities of Wageningen University & Research (51°59'41.72'N, 5°39'17.89"E, WGS-1984) and covers approximately 0.3 ha. The coordinates of the field's outer corners were recorded as Ground Control Points (GCPs) using RTK-GPS (Topcon FC-336, Japan) equipment. The field comprises of 100 squared (3 x 3 m) monoculture and 40 pairwise adjacent rectangular (3 x 1.5 m) bi-culture agricultural plots laid out in a gridded pattern, spaced 2 m and 1.5 m apart in the NE-SW and SE-NW direction, respectively. The pattern followed a randomized block design, five blocks in NW-SE direction. During the 2014 and 2015 growing season half (70) of the plots were cultivated with oat (*Avena sativa*) and the remainder with endive (*Cichorium endivia*). To enhance realism of field practices and heterogeneity, half of the replicates of the plots were rotated across both years such that oat was grown after a previous main crop of oat or endive. Furthermore, between the two main crop seasons, seven different cover crop treatments were established: plots were left fallow or were sown with *Lolium perenne* (Lp, English ryegrass), *Vicia sativa* (Vs, common vetch), *Raphanus sativus* (Rs, radish) and *Trifolium repens* (Tr, white clover) in monoculture or as species mixture of Lp+Tr or Rs+Vs. The cover crop treatments were applied randomly within each field block and in both rectangles making up the square plot (fallow and monocultures) or at either of the two rectangles making up the plot (mixtures) (Fig. 1). Except for *L. perenne* all cover crops originate from a plant family different from *A. sativa's,* hereby allowing assessment of whether such biological (dis)similarities exert influence on performance of the following plant species.

## 2.2 Field data collection and traits derivation

Samples for plant trait analysis were acquired at the grain filling stage of the 2015 growing season in each plot cultivated with *A. sativa* near-concurrent with the UAV flight on July 1[st] (Fig. 2). It was assumed that traits are homogenously distributed within plots, and the samples are thus considered representative of the entire plot. On June 30[th] mean above ground plant height (cm) was determined by means of a ruler on individual plants measured from soil level to top of the plant at four or two locations in monoculture and bi-culture plots, respectively. SPAD readings were collected on July 1[st] using a Minolta SPAD-502 meter. Measurements were taken from the top three leaves of a single plant at four locations in both monoculture and bi-culture plots. The SPAD values were then averaged and converted to leaf chlorophyll content (LCC) (g m$^{-2}$ projected leaf area) using the regression functions derived by Uddling et al. (2007) for wheat crops. Fresh biomass (kg m$^{-2}$) was recorded on July 2[nd] by clipping and weighing all above-ground vegetation using a 0.25 x 0.25 cm quadrant once in each plot. It was then oven dried at 70$^{o}$C for 48 hours to retrieve dry biomass (g m$^{-2}$). Hereafter, the nitrogen

(N) concentration (% g$^{-1}$ dry weight) was analyzed on homogenized and ground samples. Per sample we used a subsample of 150 mg, this was weighed in tin cups and then analyzed in an automated NA1500 CN elemental analyzer (Carlo Erba – Thermo Fisher Scientific). Plant N content (g m$^{-2}$) was quantified by multiplying the plant N concentration with the plant dry weight biomass.

**2.3 UAV data collection and processing**

Airborne imagery of the study site was acquired on July 1$^{st}$ 2015 on a cloud free day by the Unmanned Aerial Remote Sensing Facility (UARSF) of Wageningen University. The flight was conducted using an octocopter UAV (Aerialtronics Altura AT8) carrying a custom-built Hyperspectral Mapping System (HYMSY) sensing platform (Suomalainen et al., 2014), consisting of a pushbroom spectrometer (Specim ImSpector V10 2/3" + PhotoFocus SM2-D1312), a 16MPix consumer RGB frame camera (Panasonic GX1 + 14mm pancake lens), and a GPS-Inertial Navigation System (XSens MTi-G-700). Using the HYMSY setup three products were derived, namely (i) a RGB orthomosaic, (ii) a Digital Surface Model (DSM), and (iii) a Hyperspectral Data Cube (HDC). The imaging spectrometer data were acquired across two parallel flight lines with 80 % side overlap at a speed of 4 m/s and an altitude of 60 m. Shortly prior to take-off, the spectrometer was field calibrated for incident irradiance by taking measurement of a 25 % Spectralon reference panel. The resulting imagery was radiometrically calibrated and geometrically corrected according to the procedures presented in Suomalainen et al. (2014). As the geometrical accuracy of HDC was found to be inadequate, an additional georeferencing was performed using Esri ArcMap 10.3.1 and a custom made reference map of the study site's layout was created to further minimize geometric irregularities. Once processed, the HDC data comprised of reflectance values across 94 contiguous bands from 450 to 915 nm with 5 nm intervals, a spectral resolution of 30 nm, and a Ground Sampling Distance (GSD) of 0.14 m.

Next a Crop Surface Model (CSM) was produced. Firstly, the DSM was derived from the RGB images using Agisoft PhotoScan Pro (v1.1.2) at a pixel size of 2.9 cm. Then, the areas between plots in the DSM were interpolated to retrieve an approximated ground surface Digital Elevation Model (DEM) also in crop covered areas. The DEM was subsequently differenced with the DSM to produce the CSM depicting within plot variation of estimated plant height (Fig. 3).

To extract imaging spectrometer and canopy height data for each experimental plot, Region of Interest (RoI) polygons were manually drawn for each plot. A 30 cm border was excluded from ROIs to retrieve average plot reflectance spectra from the HDC and height from the CSM while minimizing edge effects. Inspection of the RGB orthomosaic identified significant within-plot physical heterogeneity in thirteen individual plots. We believe this was caused by accumulation of pathogens and/or nematodes under a distinct treatment. Due to the resultant conflict with the assumption of plot homogeneity required for the analysis (see 2.2), these plots were removed from the final analysis. Incorrect preprocessing of the data resulted in the cut-off of one additional plot, lowering the number of analysis objects to 41 monoculture and 15 bi-culture plots.

**2.4 Data analysis**

The resulting dataset, consisting of 56 plots, was randomly split in a calibration (50 %) and validation (50 %) set, provided that all cover crop treatments were equally divided. For both sets, the Pearson product-moment

correlation coefficient was calculated to determine the relations between the four selected plant traits, while also the correlation of these traits with the height determined from the CSM was evaluated. Next, the calibration set was used to establish relationships between the airborne UAV data and *in situ* measured crop traits through: i) (univariate) linear regression of a selection of existing vegetation indices (VIs) based on their demonstrated success for correlating well with the traits presented here (Table 1); ii) derivation of alternative two-band VIs (Aasen et al., 2014); and iii) adopting full-spectrum partial least square (PLS) regression. For derivation of alternative VIs (ii), an optimization algorithm was written in *R* to generate correlation matrices considering all possible (8,836) band combinations in simple ratio (SR), normalized difference (ND) and simple difference (SD) vegetation indices (Aasen et al., 2014). PLS regression followed earlier described procedures (Hansen and Schjoerring, 2003; Nguyen and Lee 2006; Cho et al., 2007; Abdi, 2010; Yu et al., 2014;). The optimum number of latent variables to include in PLS models was based on the minimum predicted residual sum of squares (PRESS) during leave-one-out cross validation (LOOCV), in agreement with Nguyen and Lee (2006). The performance of calibrated models was assessed using the coefficient of determination ($R^2$), i.e. an indication of how adequately dependent variables (traits) can be explained by the model (Blackburn, 1998; Maindonald and Braun, 2010; Kooistra et al., 2014).

The prediction ability of the best performing calibrated models per trait was subsequently evaluated on the independent validation dataset. Prediction precision and accuracy of the models was assessed by means of the coefficient of determination ($R^2$), Root Mean Square Error (RMSE) and the normalized RMSE (NRMSE (%)). Lower values for the latter two statistics and higher $R^2$ indicate enhanced predictive capabilities and model adequacy, respectively (Nguyen and Lee, 2006; Reddy, 2011; Li et al., 2014). The ability to discriminate between cover crop treatment effects was evaluated by regression of mean trait values per treatment measured *in situ* in the field (height, fresh biomass, LCC) or in the lab (N content) with those predicted by means of the remotely sensed UAV data and the single best performing model found during calibration. The accuracy was determined by the normalized RMSE (NRMSE) resulting from the regression analyses for each trait separately. Statistical differences between plant treatments for *in situ* quantified plant traits and for the predicted plant trait values, were tested in SPSS using ANOVA for the response variables oat plant height, fresh biomass and chlorophyll content with cover crop treatment as predicting variable. Block was initially included in the models but as it was not significant it was not included in the final models. Plant height was Ln transformed and mean fresh biomass was square root transformed prior to the analyses in order to meet the assumptions for parametric testing. Differences between treatments were analyzed using Tukey post-hoc test. Plant treatment effects on oat plant N content were analyzed using ANOVA for the predicted values and a non-parametric Median test for the *in situ* quantified data, as assumptions for parametric testing were not met despite data transformations, and differences between treatment levels were tested using two-sample Kolmogorov-Smirnov tests.

## 3. Results

### 3.1 Plant traits variation

The full-factorial randomized field experiment with different treatments of preceding plant species resulted in various degrees of variation in the traits of the subsequently grown test species *A. sativa* (Table 2). The LCC and plant N content of *A. sativa* displayed the largest dispersion, considering their associated coefficients of variation

($CV_{LCC-N}$= 0.35). Differences in mean trait values varied only marginally (< 3%) between the calibration and validation dataset. Minimum and maximum values were more strongly deviating for some traits, particularly for fresh biomass and N content, where the extreme low and/or high values in the validation set exceed the calibration data. All traits were positively correlated with each other in both the calibration and validation datasets. The strongest correlations were observed between plant height and N content and between fresh biomass and N content ($r$ between 0.8 and 0.9), the correlation between fresh biomass and LCC was notably lower ($r$ between 0.37 and 0.58).

**3.2 Effect of preceding plant treatments on succeeding plant canopy reflectance**

Spectral signatures for different plant treatments displayed deviations along a vertical rather than a horizontal axis, i.e. the relative shape of signatures was largely identical for all treatments (Fig. 4). In the visible spectrum, the highest (6 %) and lowest (3 %) reflectance were recorded at 555 nm and 675 nm, i.e. the chlorophyll absorption minimum and maximum, respectively (Brodge and Leblanc, 2000; Haboudane et al., 2002; Vincini et al., 2007). Beyond the chlorophyll post-maxima (± 700 nm), reflectance greatly increased over red-edge wavelengths and up to sevenfold of the maximum visible reflectance (> 45%) in the near-infrared spectral region (> 750 nm) due to increasing crop biomass (Nguyen and Lee, 2006). Absolute variations were also most strongly pronounced at the latter wavelengths. Plots left fallow as pre-treatment or those that were cultivated with *L. perenne*, *T. repens* or their combination (Lp+Tr) consistently exhibited the lowest canopy reflectance of *A. sativus* in the near-infrared. In contrast, near-infrared reflectance was highest for plots previously cultivated by *R. sativus*, *V. sativa* or a combination of these two cover crop species. *In situ* sampling also recorded the highest values for fresh biomass for these treatments (Fig. 4).

**3.3 Univariate trait correlation with crop surface model (CSM) height**

CSM height was positively correlated to all crop traits, particularly for validation plots (Table 3). In general, the observed interdependencies confirmed the associated relationships between vegetation height and variables such as growth rate, biomass and plant fertility/health (e.g., Cornelissen et al., 2003; Tilly al. 2014). Strongest correlations were observed for *in situ* measured crop height, indicated by correlation coefficients of 0.85 and 0.91 for calibration and validation data, respectively. Furthermore, relative variations in CSM height were also significantly (p < 0.001) related to *in situ* measured height discrepancies for different treatments ($R^2 \approx 0.95$, NRMSE ≈ 27.4 %). The CSM, however, exhibited some bias and underestimated *in situ* measurements by 20cm on average. The other plant traits, i.e. N content ($r \approx 0.69/0.73$), LCC ($r \approx 0.67/0.79$) and fresh biomass ($r \approx 0.62/0.74$), displayed slightly lower correlation coefficients.

**3.4 Calibration**

**3.4.1 Relationship between existing vegetation indices (VIs) and crop traits**

*In situ* measurements were linearly regressed with a selection of well-established VIs (Table 1) based on the best matching bands from the HDC, the main product of hyperspectral mapping system. Regression analysis yielded highly varying $R^2$ values for different combinations of traits and existing VIs (Table 4), although indices yielding comparatively high coefficients of determination in relation to a distinct trait were generally found to also be rather strongly correlated to multiple other traits. The relationship between *in situ* measured crop traits

and VIs was strongest for the REP and MTCI indices, particularly for height ($R^2 \approx 0.69$), N content ($R^2 \approx 0.59$) (Fig. 5a), LCC ($R^2 \approx 0.58$) (Fig. 5b) and, albeit to a lesser degree, for fresh biomass ($R^2 \approx 0.25$). The performance of these indices was closely followed by some of the evaluated two-band indices, NDVI in particular (Table 4). Exponential fitting improved $R^2$ values for LCC in particular, albeit marginally, by 0.06 at most in some instances. This relatively minor improvement invoked by exponential fitting, compared to findings in previous studies, may be the result of the relatively limited range of LCC values.

In agreement with the wavelength dependency of REP (670 nm, 700 nm, 740 nm, 780 nm) and MTCI (680 nm, 710 nm, 755 nm), the best performing two-band indices recurrently exploit the near-infrared (> 750 nm) and the far red (± 710 nm), the red-edge (between 710 nm and 750 nm) and the far red, or solely the red-edge. Contrastingly, indices that performed relatively weak appeared to be primarily based on wavelengths in the visible part of the spectrum, particularly in the green (± 550 nm) and the blue. Soil background noise mitigating indices (i.e. TACRI/OSAVI and MACRI/OSAVI) did not enhance performance compared to their non-adjusted counterparts. This may be attributed to the advanced stage of the crops studied and the resulting dense canopy cover, rendering the appearance and influence of soil background largely absent (Thenkabail et al., 2000).

**3.4.2 Selecting alternative VIs for estimating plant traits**

In order to explore the applicability of alternative band combinations, plant traits were linearly regressed against all possible simple ratio (SR, $\lambda1/\lambda2$), normalized difference (ND, ($\lambda2 - \lambda1$)/($\lambda2+\lambda1$)) and simple difference (SD, $\lambda1-\lambda2$) vegetation indices using the 94 wavelengths and the associated measured reflectance (Aasen et al., 2014). The resulting range of $R^2$ values displayed stronger relationships for all traits compared to the evaluated existing two-band indices. Compared to MTCI and REP, increments in coefficients of determination for height, fresh biomass and N content were only observed for SD indices. Furthermore, new SD indices exhibited higher $R^2$ values than SR and NDVI indices for all traits except height, although variation in maximum coefficients of determination for different index formulations was small (< 0.03). Likewise, the highest $R^2$ values observed for existing or new indices varied marginally (between 0.03 and 0.1).

The hotspots identified for SRs largely aligned with those found for NDVIs, and to a smaller degree with optimized SD indices (Table 5). In accordance with the earlier findings for existing indices (Table 4), the best performance was observed for indices borrowing from the red-edge (> 725 nm) or near-infrared spectral (> 750 nm) region. An alternative near-infrared oriented SD index (875 nm-915 nm) produced a marginally improved $R^2$ for fresh biomass. Relatedly, in contrast to findings in a variety of previous studies, indices fully oriented at the visible spectrum or, alternatively, indices exploiting the red-edge/near-infrared and visible wavelengths, were found to be less strongly related to measured crop traits. Following from the considerable correlations between trait pairs, hotspots for different combinations of indices and crop traits were largely overlapping, although it was marginally extended to shorter red-edge wavelengths (> 710 nm) for LCC and N content. Furthermore, exploiting longer near-infrared wavelengths invoked relatively faster lowering of $R^2$ values for LCC and fresh biomass compared to height and N content.

### 3.4.3 Partial Least Square Regression for estimating plant traits

Finally, spectra were related to plant traits employing two partial least square (PLS) regression models. The first model (PLS1) incorporated all mean plot reflectance measurements in the 450 - 915 nm range, the second (PLS2) included plot-wise height measurements derived from the CSM as an additional explanatory variable. The optimum number of latent variables (NLV) in the PLS1 models ranged from 1 and 3 for fresh biomass and height to 5 and 11 for LCC and N content, respectively. The NLV in PLS2 models for height and N content changed to 5 and 2, respectively. The model precision and accuracy was highest for height, LCC and N content, indicated by the $R^2$ (coefficients of determination) and NRMSE (Normalized Root Mean Square Error), respectively (Table 6). Compared to the best performing existing or new indices, PLS1 models improved $R^2$ values by 0.05, 0,18 and 0.32 for height, LCC and N content, respectively. The PLS2 model only produced higher $R^2$ values for height (+ 0.18) and fresh biomass (+ 0.12).

The factor loadings indicated the relative importance of explanatory variables for the construction of each LV, i.e. higher loadings attribute comparatively more influence (Hansen and Schjoerring, 2003; Nguyen and Lee, 2006). It was observed for all traits in PLS1 models that the first loading weights allocate significant leverage to longer red-edge and near-infrared wavelengths in particular. High loading weights for the second component were recorded at 710 nm for LCC, and at 560 nm in the green peak for height and N content. In all PLS2 models, CSM height was accredited with the highest loading score for all traits. Consequently, the PLS2 model for fresh biomass (NLV = 1) was merely a linear function of CSM height rather than reflectance.

### 3.5 Model validation

#### 3.5.1 Model prediction accuracy

The independent validation dataset was employed to assess plant trait prediction accuracies of previously calibrated models, including the three best performing existing indices, one of each optimized new index and the best of two PLS models for each trait (Table 7). The highest prediction accuracies were obtained for crop height (NRMSE= 5.12 %, $R^2$= 0.79) and LCC (NRMSE= 14.5 %, $R^2$= 0.79), based on existing indices MTCI and REP, respectively. Differences in highest prediction accuracies for fresh biomass (NRMSE= 20.78 %, $R^2$= 0.56) and N content (NRMSE= 21.6 %, $R^2$= 0.68) were negligible, although the model precision for the latter trait was higher. Furthermore, the best results for N content were provided by the optimized SD index, whereas the second PLS model type (i.e. a linear function of CSM crop height) delivered the highest accuracies for fresh biomass. These results, in particular of the NRMSE scores, show that optical UAV based remote sensing can be an applicable means for in field quantification of biophysical and biochemical oat plant characteristics to different degrees.

#### 3.5.2 Plant-soil feedback treatment discrimination

The plant legacies resulted in significant differences ($\alpha = 0.05$) in the traits of the following grown oat crop, as indicated by the results of the analyses of variance for the different traits. We found significant treatment effects in oat plant height ($F_{6,21}$= 11.99, p< 0.001), fresh biomass ($F_{6,21}$= 4.93, p< 0.01), chlorophyll content ($F_{6,21}$= 11.10, p< 0.001) and N content ($\chi^2$= 15.2, p< 0.05) on *in situ* measured plant traits (Fig. 6a-d). Similar results were found when using the predicted plant trait values from the remote sensing data to test the soil legacy

effects: we found significant effects of plant legacies on oat plant height ($F_{6,21}= 18.05$, $p< 0.001$), fresh biomass ($F_{6,21}= 24.58$, $p< 0.001$), leaf chlorophyll content ($F_{6,21}= 26.91$, $p< 0.001$) and N content ($F_{6,21}= 11.87$, $p< 0.001$) (Fig. 6e-h). Mean plant height and chlorophyll content were highest in oat growing in soil with a legacy of *V. sativa* and of *V. sativa* mixed with *R. sativus*, and were lowest in soil with a legacy of *L. perenne*. Fresh biomass and N content on the other hand appeared to be largest in oat growing in soil with a legacy of *R. sativus* or *V. sativa* in monoculture. Variations in *in situ* measured mean trait parameters were regressed with predicted mean trait values according to the best performing prediction model for each individual trait (Table 7). Using the model predicted values derived from remote sensing resulted in successful discrimination between the field treatments (Fig. 6e-h), i.e. plant legacies, for plant height (CVRMSE= 3.03 %, $R^2= 0.92$) and LCC (CVRMSE= 4.93 %, $R^2= 0.97$). The relative and absolute quantitative differences of both traits were largely aligned across different treatments ($p < 0.001$). In contrast, remotely sensed treatment effects for fresh biomass were only partially matching the *in situ* quantified values for lower biomass levels, whereas discrimination among higher biomass was largely absent or contradictory in the remotely sensed data (CVRMSE= 14.01 %, $R^2= 0.79$). Absolute and relative characterization of treatment effects for N content (CVRMSE= 16.61 %, $R^2= 0.79$) was rather successful for all treatments other than *R. sativus* (+ *V. sativa*). However, due to the removal of plots that did not comply with the required plot homogeneity (see 2.3), observations for these treatments were significantly underrepresented.

## 4. Discussion

### 4.1 Relevant wavelengths for plant trait predictions

Across all traits and analysis methodologies the red-edge and near-infrared spectral region were consistently of critical importance, in contrast to visual wavelengths. This finding is in agreement with expectations based on univariate correlations of traits over wavelengths (not shown). The red-edge slope is of particular relevance for leaf chlorophyll content (LCC) because of its enhanced sensitivity to varied and higher chlorophyll levels while circumventing saturation problems as observed in the blue and red due to vast chlorophyll induced absorption (Gitelson, 2012; Kooistra and Clevers, 2016). Reliance on wavelengths at the onset of the near-infrared follows from the gradual stabilization of reflectance beyond the red-edge, which settles at higher values for increased chlorophyll levels (Lamb et al., 2002). Largely similar spectral regions were structurally highlighted for N content, resulting from the inherent biochemical linkage between leaf N, chlorophyll molecules and photosynthetic capacity (Sellers et al., 1992; Weiss et al., 2001; Netto et al., 2005; Wu et al., 2008). Consequently, wavelengths positioned in the red-edge were found to be highly sensitive for chlorophyll absorption behavior and thus to accumulation of nitrogen (Thenkabail et al., 2012; Zhao et al., 2014). Although a direct physical relationship between plant height and reflectance is absent, alternative structural parameters (e.g. biomass and canopy densification) may serve as a proxy for the former (Wang et al., 2011). Resultantly, the employing of near-infrared wavelengths regarding plant height possibly followed from *A. sativa* growing taller, while gradually disclosing additional leaves that enhanced near-infrared scattering (Christenson et al., 2013). Relatedly the accuracy of the PLS2 model (i.e. a linear function of CSM crop height) regarding fresh biomass may be explained by comparable interdependencies between crop height and fresh biomass. The focus on adjacent red-edge and/or short near-infrared wavelengths, rather than red(-edge) oriented models, follows from

these being a better estimator of biomass in dense vegetation (e.g. full grown *A. sativa*) (Mutanga and Skidmore, 2004).

### 4.2 Plant traits and plant physiological stage

It is well-known that plant traits vary according to plant growth stage. As plants mature and start senescing, stocks of both N and biomass are gradually re-allocated to grains, hereby invoking reduced photosynthetic capacity, discoloring of leaves and exposing of other plant pigments (Peinetti et al., 2001; Murphy and Murray, 2003; Ciganda et al., 2009). Consequently, various previous studies found that estimation of N (Zhao et al., 2014), biomass (Yang and Miller, 1985) and height (Scotford and Miller, 2004) in mature vegetation is prone to larger inaccuracies compared to in vegetation in earlier growth phases. In our study measurements were obtained during grain-filling stage, so in mature plants. The loss in photosynthetic capacity in senescing plants likely partially explains the reduced importance of visible (i.e. red) wavelengths, as chlorophyll absorption at these wavelengths becomes less pronounced than generally is the case during preceding stages (Gitelson, 2012). Besides, the maximization of biomass accumulation in matured plants (Malhi et al., 2006) may explain lower $R^2$ values for fresh biomass estimations, as spectral sensitivity to biomass saturates at higher biomass levels (Goswami et al., 2015). Therefore, to evaluate the validity and robustness of the relationships found in the current study, and to explore whether different models exhibit other prediction capabilities at alternative growth stages we recommend future studies to perform observations and analyses across the plant vegetative cycle. In our study we tested a range of NDVI indices which have been reported in literature to link with one or two plant traits, because the best fitting index was not *a priory* known. Across the indices we found a lot of redundancy, however, for measurements during a different plant physiological stage some of these indices may prove to be a better fit than they appeared to be for our data.

### 4.3 Ability to discriminate plant legacy effects in soil using UAV-based sensor data

The different cover crop treatments resulted in marked differences in several plant traits of the following crop of *A. sativa*, namely plant height, fresh biomass, plant nitrogen and leaf chlorophyll content. These plant-soil feedbacks from the cover crops to the *A. sativa* crop are generated via nutrient mineralisation/immobilisation which supports/constrains plant growth and these are linked to different organic matter inputs resulting from the cover crop treatments (Hodge et al. 2000). Also the build-up of plant growth suppressing organisms can suppress plant height, biomass and nitrogen content. These effects however are more patchy/less homogeneous than plant-soil feedbacks generated via nutrient cycling. The short stature and low chlorophyll and nitrogen content in *A. sativa* grown after the cover crop treatments with the grass *L. perenne* and conversely high values for these plant traits in *A. sativa* grown after treatments with the legume *V. sativa* indicate a role for (temporal) nutrient immobilisation as *L. perenne* shoots and roots decompose and fast mineralisation of dead shoots and roots of *V. sativa* but also of the brassica species *R. sativus*. The relative poor performance of the *A. sativa* plant traits after the legume *T. repens* is likely due to the poor establishment we observed for *T. repens* and consequently low amount of organic input upon incorporation of this cover crop in the soil.

We were able to pick-up significant differences between the treatments both on the *in situ* measured and on the remote-sensing based modeled values. These results provide scope for Unmanned Aerial Vehicles (UAVs) and imaging spectroscopy as an enabling means to transfer plant-soil feedback studies and related studies on legacy

effects in soil to outdoor field environments (Fiorani and Schurr, 2013; Faye et al., 2016). To date most studies on plant legacies in soil and their impact on subsequent plant growth have been performed under controlled greenhouse conditions at small scale, yet it has been advocated that outdoor field experiments are needed to assess the magnitude and relevance of plant legacies (Kulmatiski et al., 2008; van der Putten et al., 2013). Here we propose, based on the results of our current study, that the use of UAV-based optical sensors allow for adequate field observations that will enable to complement and/or verify associate studies executed in controlled indoor environments. The use of UAVs is faster and more cost-efficient compared to conventional (i.e. hand-held) means, while limiting the intrusion of changing atmospheric conditions to affect measurements (Chapman et al., 2014). Moreover, UAVs enable operational resilience, besides adequate scaling of spatial detail and temporal revisiting times without the need for destructive sampling to measure and monitor ecological phenomena such as successional physiological vegetation processes over time (Faye et al., 2016).

## 4.4 Future improvements

Apart from differences in plant traits between plant physiological stages it also has to be noted that the quality of the predicted values is dependent on constraints invoked by the quality and quantity of ground truth data (Michaelsen et al., 1994). *In situ* sampling was conducted at diversified densities for different traits and/or for monoculture and biculture plots (see 2.2). Following from hypothesized plot trait and treatment homogeneity, samples were considered representing the remainder of plots. However, some degree of within plot heterogeneity was present. Consequently, calibration and validation of relationships between plot averaged spectra and field samples at one or a limited number of locations may have been suboptimal. To enhance the robustness of the models we therefore advise future studies to use a more extensive sampling layout such that field sample locations more accurately align with UAV spectrometer data from which data is further processed (von Bueren et al. 2014). Furthermore improvements can be made by using a different flight, performed on a subsequent day, as a validation data set to evaluate retrieval model sesitivity and by performing more flights over the growing season to capture temporal variation (Capolupo et al. 2015). The processing of the data to derive plant trait indices from the spectra collected using UAV-mounted sensors can be improved by making use of bootstrapping to find the best combinations of indices (Souza et al. 2010) and machine learning techniques based on the available spectral wavelengths (Singh et al. 2016).

## 5. Concluding remarks

Plant-soil feedback (PSF) studies gained scientific interest over the last decades, however field studies are urgently needed in order to evaluate the role of PSF processes under field conditions (Kulmatiski et al., 2008; van der Putten et al., 2013). Here we show that UAV-based hyperspectral remote sensing of plant traits enables to non-destructively quantify plant traits that respond to plant legacies in soil. This finding offers great potential to expand studies of PSF effects from the greenhouse to field settings. The plant traits that could be most accurately and precisely quantified were plant height and leaf chlorophyll content. The non-destructive nature of the measurements, after thorough parameterization, furthermore enables studying PSF effects at field scale at relevant spatial-temporal resolutions, this in turn will facilitate the elucidation of the underlying mechanisms.

## 6. Data availability

The data will be made available via publicly accessible data repository Dryad and upon request to the authors.

## 7. Acknowledgements

We thank the staff from Unifarm and Irene Garcia Gonzalez and Dominika Piwcewicz for help with the field work. This work was supported by an NWO-ALW VIDI to GBDD (grant nr 864.11.003).

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

**Table 1.** Overview of existing vegetation indices that were evaluated in this study for retrieving plant traits from optical remote sensing images. A division is made in two-bands indices based on simple ratios and NDVI, and other indices using more than two bands. The index description includes their formulation and the original references.

| Index name | Formulation | Reference |
|---|---|---|
| **Simple Ratios** | | |
| **SR_a** | $\dfrac{R734}{R629}$ | *Yu et al. (2012)* |
| **SR_b** | $\dfrac{R780}{R710}$-$1$ | *Gitselson (2003),* |
| **SR_c** | $\dfrac{R780}{R550}$-$1$ | *Clevers & Kooistra (2012);* |
| **SR_d** | $\dfrac{R760}{R550}$ | *Zhao et al. (2014)* |
| **SR_e** | $\dfrac{R706}{R755}$ | *Mutanga & Skidmore (2004)* |
| **MSR** | $\dfrac{(R750/R705) - 1}{\sqrt{(R750/R705) + 1}}$ | *Wu et al. (2008)* |
| **NDVIs** | | |
| **NDVI_a** | $\dfrac{R689 - R521}{R689 + R521}$ | |
| **NDVI_b** | $\dfrac{R584 - R471}{R584 + R471}$ | |
| **NDVI_c** | $\dfrac{R732 - R717}{R732 + R717}$ | *Hansen & Schjoerring (2003)* |
| **NDVI_d** | $\dfrac{R750 - R734}{R750 + R734}$ | |
| **NDVI_e** | $\dfrac{R770 - R717}{R770 + R717}$ | |
| **NDVI _f** | $\dfrac{R820 - R720}{R820 + R720}$ | *Thenkabail et al. (2000)* |
| **NDVI_g** | $\dfrac{R750 - R705}{R750 + R705}$ | *Wu et al. (2008)* |
| **NDVI_h** | $\dfrac{R740 - R667}{R740 + R667}$ | *Yu et al. (2012)* |
| **NDVI_i (NDRE)** | $\dfrac{R780 - R710}{R780 + R710}$ | *Kooistra et al. (2014)* |
| **NDVI_j** | $\dfrac{R760 - R550}{R760 + R550}$ | *Zhao et al. (2014)* |
| **NDVI_k** | $\dfrac{R750 - R710}{R750 + R710}$ | *Wu et al. (2009)* |

**Other Indices**

| | | |
|---|---|---|
| **REP_a** | $700 + 45 * \dfrac{Rre - R700}{R740 - R700} \qquad Rre = \dfrac{R670 + R780}{2}$ | *Cho et al. (2007),* |
| **MCARI_a** | $[(R750 - R705) - 0.2(R750 - R550)](\dfrac{R750}{R705})$ | *Wu et al. (2008)* |
| **MCARI_b** | $[(R750 - R710) - 0.2(R750 - R550)](\dfrac{R750}{R710})$ | *Wu et al. (2009)* |
| **TCARI/OSAVI** | $\dfrac{3[(R750 - R705) - 0.2(R750 - R550)(R750/R705)]}{(1 + 0.16)(R750 - R705)/(R750 + R705 + 0.16)}$ | *Wu et al. (2008)* |
| **MCARI/OSAVI** | $\dfrac{[(R750 - R705) - 0.2(R750 - R550)](R750/R705)}{(1 + 0.16)(R750 - 705)/(R750 + R705 + 0.16)}$ | *Wu et al. (2008)* |
| **MTCI** | $\dfrac{R754 - R709}{R709 - R681}$ | *Tian et al. (2011),* |
| **TGI** | $-0.5[190(R670 - R550) - 120(R670 - R480)]$ | *Hunt Jr. et al. (2013)* |
| **MCARI/MTVI2** | $\dfrac{(R700 - R670 - 0.2(R700 - R550)) * (R700/R670)}{1.5(1.2(R800 - R550) - 2.5(R670 - R550))/\sqrt{((2R800 + 1)^2 - (6 * R - 5 * \sqrt{(R670)}) - 0.5}}$ | *Tian et al. (2011),* *Chen et al. (2010)* |

**Table 2.** Summary of descriptive statistics for all plant traits measured in the field. LCC= leaf chlorophyll content, SD= standard deviation, CV= coefficient of variation.

| Plant trait | Unit | Calibration set (n= 28) | | | | | Validation set (n= 28) | | | | |
|---|---|---|---|---|---|---|---|---|---|---|---|
| | | Mean | SD | CV | Min | Max | Mean | SD | CV | Min | Max |
| **Height** | cm | 91.26 | 10.04 | 0.11 | 72.50 | 112.50 | 89.67 | 9.92 | 0.11 | 72.50 | 108.88 |
| **Fresh biomass** | kg m$^{-2}$ | 3.54 | 0.86 | 0.24 | 2.19 | 5.41 | 3.47 | 1.04 | 0.30 | 1.67 | 5.77 |
| **N content** | g m$^{-2}$ | 8.98 | 2.91 | 0.32 | 5.14 | 17.85 | 9.00 | 3.27 | 0.36 | 3.87 | 15.23 |
| **LCC** | g m$^{-2}$ | 0.76 | 0.28 | 0.37 | 0.38 | 1.38 | 0.72 | 0.23 | 0.32 | 0.41 | 1.27 |

**Table 3.** Correlation coefficients (r) based on a linear regression between average plot height derived from the Crop Surface Model and each selected plant trait for the calibration (n= 28) and validation set (n= 28), LCC= leaf chlorophyll content.

| Plant trait | Calibration | Validation |
|:---:|:---:|:---:|
| Height | 0.85 | 0.91 |
| Fresh biomass | 0.62 | 0.74 |
| N content | 0.69 | 0.73 |
| LCC | 0.67 | 0.79 |

**Table 4.** Coefficients of determination ($R^2$) based on a linear regression for the calibration set between existing vegetation indices (VI) and the studied plant traits, LCC= leaf chlorophyll content. The three models producing the highest coefficients for each trait are displayed in bold.

| Vegetation index | Height | Fresh biomass | N content | LCC |
|---|---|---|---|---|
| SR_a | 0.050 | 0.032 | 0.078 | 0.117 |
| SR_b | 0.472 | 0.178 | 0.450 | 0.488 |
| SR_c | 0.147 | 0.067 | 0.175 | 0.231 |
| SR_d | 0.120 | 0.057 | 0.149 | 0.207 |
| SR_e | 0.337 | 0.132 | 0.326 | 0.412 |
| MSR | 0.335 | 0.130 | 0.332 | 0.398 |
| NDVI_a | 0.278 | 0.078 | 0.169 | 0.142 |
| NDVI_b | 0.354 | 0.105 | 0.215 | 0.220 |
| NDVI_c | 0.338 | 0.134 | 0.334 | 0.410 |
| NDVI_d | 0.566 | **0.197** | **0.505** | **0.546** |
| NDVI_e | 0.492 | 0.184 | 0.454 | 0.525 |
| NDVI_f | 0.533 | 0.191 | 0.477 | 0.514 |
| NDVI_g | 0.327 | 0.128 | 0.321 | 0.400 |
| NDVI_h | 0.039 | 0.024 | 0.063 | 0.119 |
| NDVI_i | 0.449 | 0.170 | 0.420 | 0.493 |
| NDVI_j | 0.125 | 0.061 | 0.153 | 0.227 |
| NDVI_k | 0.380 | 0.145 | 0.364 | 0.440 |
| REP | **0.698** | **0.245** | **0.580** | **0.573** |
| MCARI_a | 0.429 | 0.158 | 0.414 | 0.475 |
| MCARI_b | 0.477 | 0.174 | 0.454 | 0.510 |
| TCARI/OSAVI | 0.195 | 0.073 | 0.207 | 0.263 |
| MCARI/OSAVI | 0.195 | 0.073 | 0.207 | 0.263 |
| MTCI | **0.679** | **0.245** | **0.599** | **0.583** |
| TGI | 0.248 | 0.090 | 0.194 | 0.222 |
| MCARI/MTVI2 | **0.655** | 0.183 | 0.459 | 0.463 |

**Table 5.** Generated optimized indices for different plant traits and their wavelength dependency

| Simple Ratio | λ1 (nm) | λ2 (nm) | Normalized Difference | λ1 (nm) | λ2 (nm) | Simple Difference | λ1 (nm) | λ2 (nm) |
|---|---|---|---|---|---|---|---|---|
| SR_i | 795 | 755 | NDVI_i | 795 | 755 | SD_i | 785 | 760 |
| SR_ii | 790 | 755 | NDVI_ii | 790 | 755 | SD_ii | 875 | 915 |
| SR_iii | 790 | 745 | NDVI_iii | 790 | 745 | SD_iii | 780 | 760 |
| SR_iv | 760 | 740 | NDVI_iv | 760 | 740 | SD_iv | 780 | 765 |
| | | | | | | SD_v | 760 | 740 |

**Table 6.** Statistical parameters of the PLS model calibration for the four selected plant traits, LCC= leaf chlorophyll content.

| Plant trait | Number of latent variables | $R^2$ (cross validation) | RMSEP | NRMSE (%) | $R^2$ (fitted values) |
|---|---|---|---|---|---|
| **Height (PLS1)** | 3 | 0.65 | 5.909 | 6.47% | 0.75 |
| **Height (PLS2)** | 5 | 0.81 | 4.377 | 4.80% | 0.88 |
| **Fresh biomass (PLS1)** | 1 | 0.06 | 0.8352 | 23.57% | 0.20 |
| **Fresh biomass (PLS2)** | 1 | 0.29 | 0.7268 | 20.51% | 0.39 |
| **N content (PLS1)** | 11 | 0.49 | 2.054 | 22.88% | 0.93 |
| **N content (PLS2)** | 2 | 0.47 | 2.127 | 23.69% | 0.58 |
| **LCC (PLS1)** | 5 | 0.63 | 0.1681 | 22.43% | 0.79 |
| **LCC (PLS2)** | 5 | 0.62 | 0.1709 | 22.81% | 0.76 |

$R^2$ = coefficient of determination, RMSEP = Root Mean Square Error of Prediction, NRMSE = Normalized Root Mean Square Error).

**Table 7.** Overview of validation statistics for the best selected existing and new indices and both PLS models for the different plant traits, LCC= leaf chlorophyll content. All results presented are significant at < 0.0001 probability level, unless the asterisks indicate otherwise (*** < 0.001, ** < 0.01, * < 0.1). For each trait the model with the highest predictive accuracy is displayed in bold. Abbreviations: RMSE = Root Mean Square Error; NRM = Normalized Root Mean Square Error; and $R^2$ = coefficient of determination. Grey cells indicate indices that were not validated as only the best performing indices per trait in the calibration were selected for validation.

| Index | Height | | | Fresh biomass | | | N content | | | LCC | | |
|---|---|---|---|---|---|---|---|---|---|---|---|---|
| **Existing indices** | RMSE | NRM | R² | RMSE | NRM | R² | RMSE | NRM | R² | RMSE | NRM | R² |
| NDVI_d | | | | 0.82 | 23.55% | 0.437*** | 2.26 | 25.11% | 0.53 | 0.15 | 21.18% | 0.61 |
| NDVI_f | | | | | | | | | | | | |
| REP | 4.70 | 5.24% | 0.78 | 0.794 | 22.89% | 0.494 | 2.04 | 22.61% | 0.63 | **0.11** | **14.50%** | **0.794** |
| MTCI | **4.59** | **5.12%** | **0.79** | 0.797 | 22.96% | 0.472 | 2.12 | 23.58% | 0.585 | 0.13 | 17.73% | 0.71 |
| MCARI/MTVI2 | 6.984 | 7.79% | 0.565 | | | | | | | | | |
| | | | | | | | | | | | | |
| *New indices* | *RMSE* | *NRM* | *R²* | *RMSE* | *NRM* | *R²* | *RMSE* | *NRM* | *R²* | *RMSE* | *NRM* | *R²* |
| SR_i | 5.19 | 5.79% | 0.74 | | | | | | | | | |
| SR_ii | | | | 0.83 | 23.91% | 0.455 | | | | | | |
| SR_iii | | | | | | | 2.06 | 22.83% | 0.628 | | | |
| SR_iv | | | | | | | | | | 0.12 | 17.16% | 0.728 |
| NDVI_i | 5.16 | 5.75% | 0.75 | | | | | | | | | |
| NDVI_ii | | | | 0.828 | 23.80% | 0.458 | | | | | | |
| NDVI_iii | | | | | | | 2.05 | 22.78% | 0.629 | | | |
| NDVI_iv | | | | | | | | | | 0.13 | 17.36% | 0.725 |
| SD_i | 4.81 | 5.37% | 0.77 | | | | | | | | | |
| SD_ii | | | | 0.741 | 21.37% | 0.56 | | | | | | |
| SD_iii | | | | 0.78 | 22.52% | 0.56 | | | | | | |
| SD_iv | | | | | | | **1.94** | **21.60%** | **0.68** | | | |
| SD_v | | | | | | | | | | 0.16 | 21.72% | 0.61 |
| | | | | | | | | | | | | |
| **PLS models** | RMSE | NRM | R² | RMSE | NRM | R² | RMSE | NRM | R² | RMSE | NRM | R² |
| PLS 1 | 4.84 | 5.39% | 0.78 | 0.77 | 22.31% | 0.50 | 3.81 | 42.34% | 0.242* | 0.17 | 23.82% | 0.57 |
| PLS 2 | 5.30 | 5.91% | 0.74 | **0.72** | **20.78%** | **0.56** | 2.05 | 22.82% | 0.62 | 0.16 | 21.63% | 0.64 |

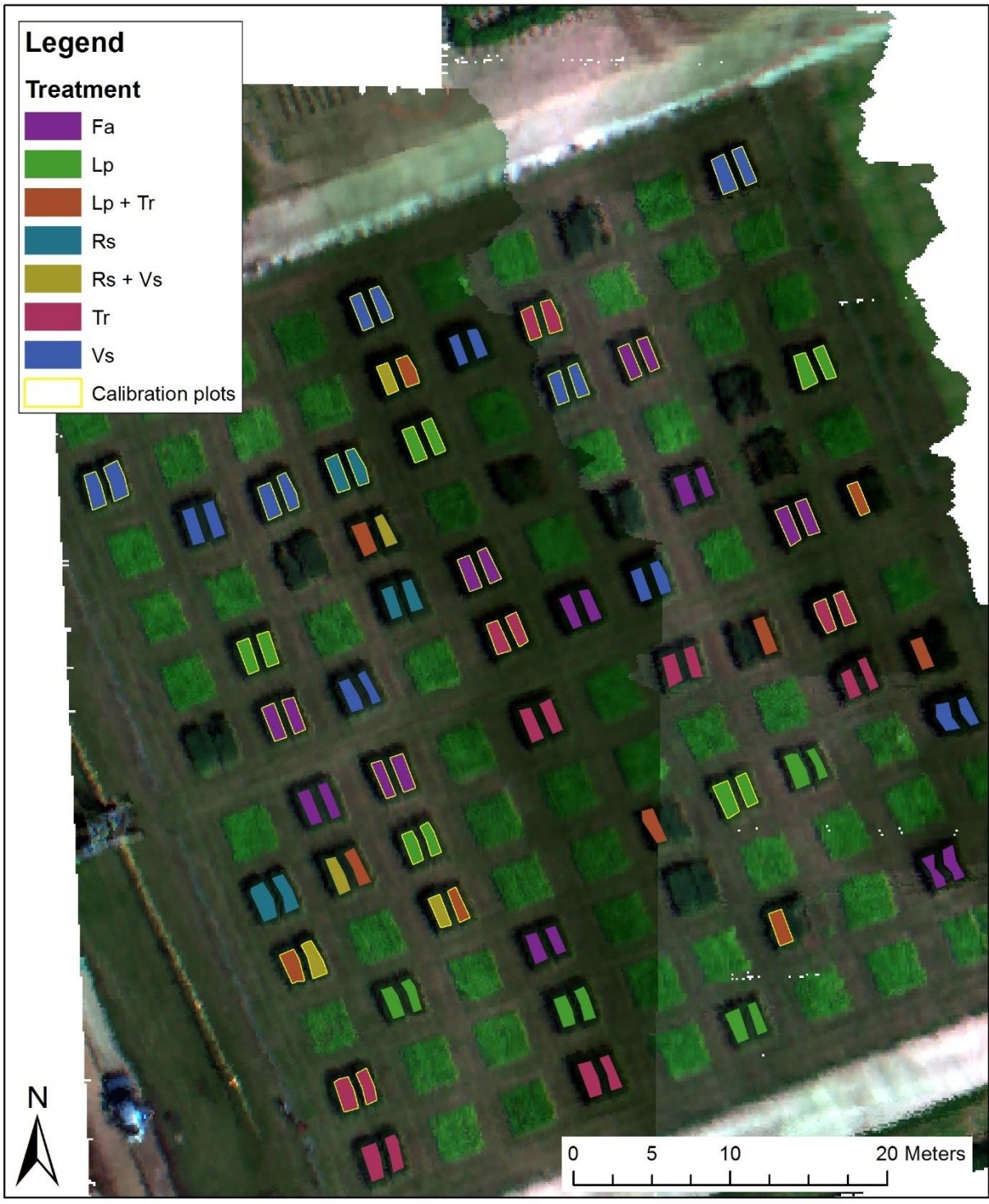

**Figure 1.** The experimental field as imaged from the Hyperspectral Data Cube (HDC) acquired on July 1, 2015 represented as true color RGB image, regions of interest (ROIs) for *A. sativa* (oat) plots and plot wise treatments. Plant legacy treatments are: Fa= fallow, Lp= *Lolium perenne*, Rs= *Raphanus sativus*, Tr= *Trifolium repens*, Vs= *Vicia sativa*, Lp+Tr= *L. perenne + T. repens*, Rs+Vs= *R. sativus + V. sativa*.

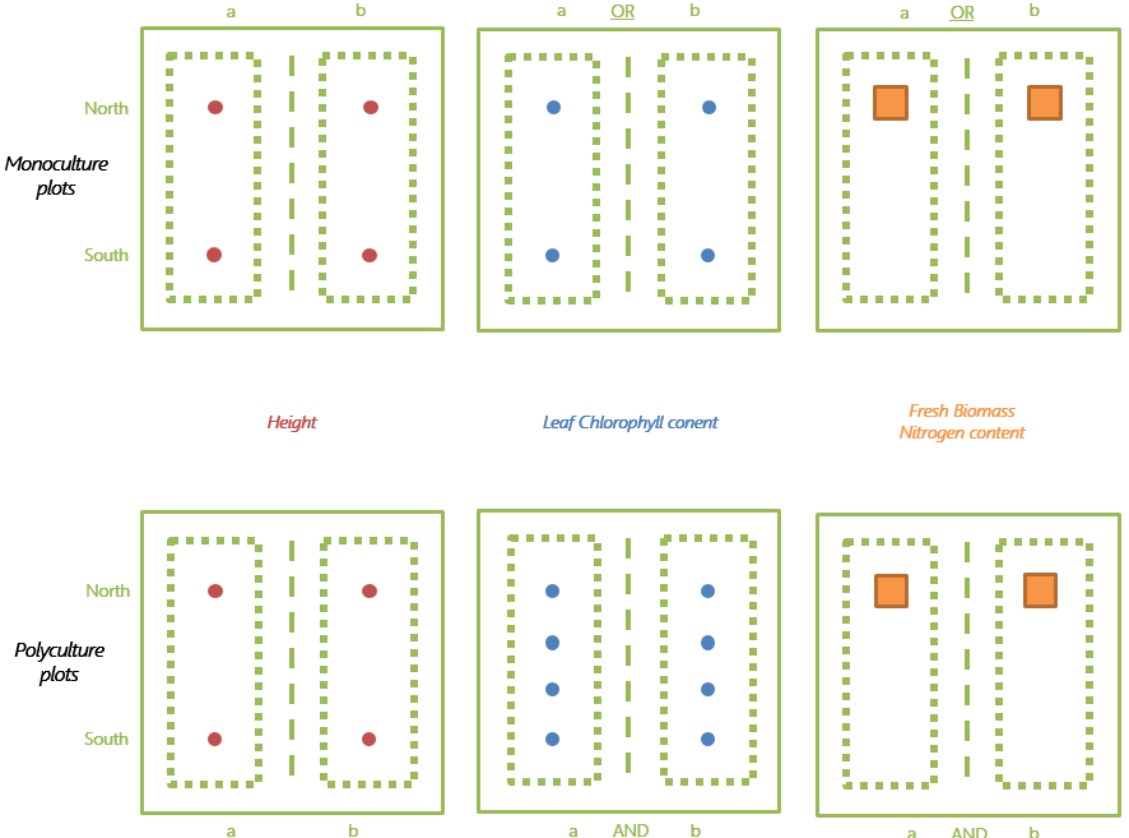

**Figure 2.** Schematic overview of individual plots and the approximated location at which samples for plant traits were collected.

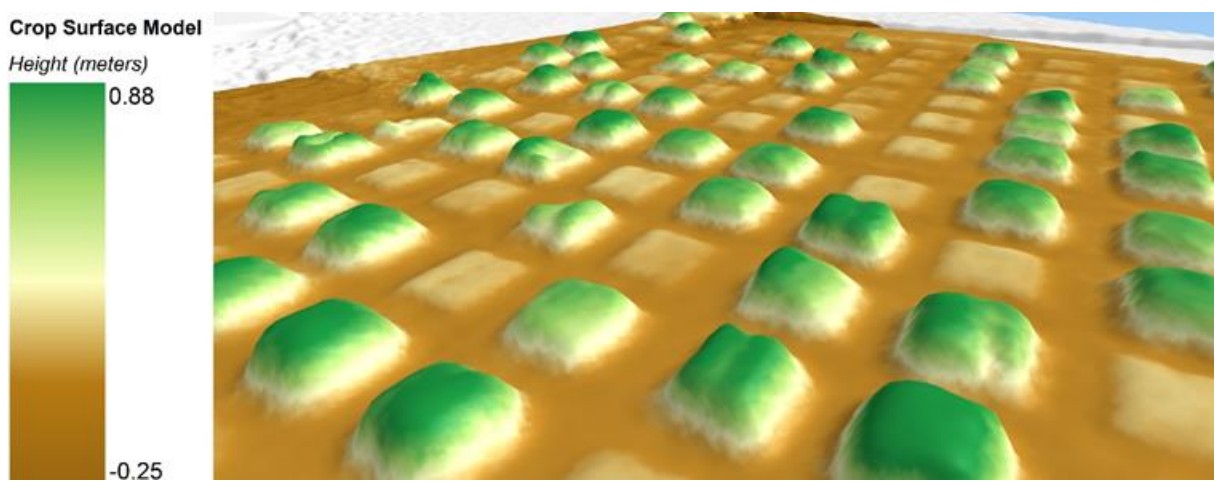

 **Figure 3.** Visual threedimensional representation of the crop surface model (DSM), upon differencing of the original digital surface model (DSM) and a secondary model approximating the ground surface Digital Elevation Model (DEM) of the study area. Vegetation height is illustrated from low to heigh by change in color from yellow, to orange, to red, to dark red. An absolute scale is not provided as the illustration is an oblique view on the CSM and not a planar two-dimensional figure.

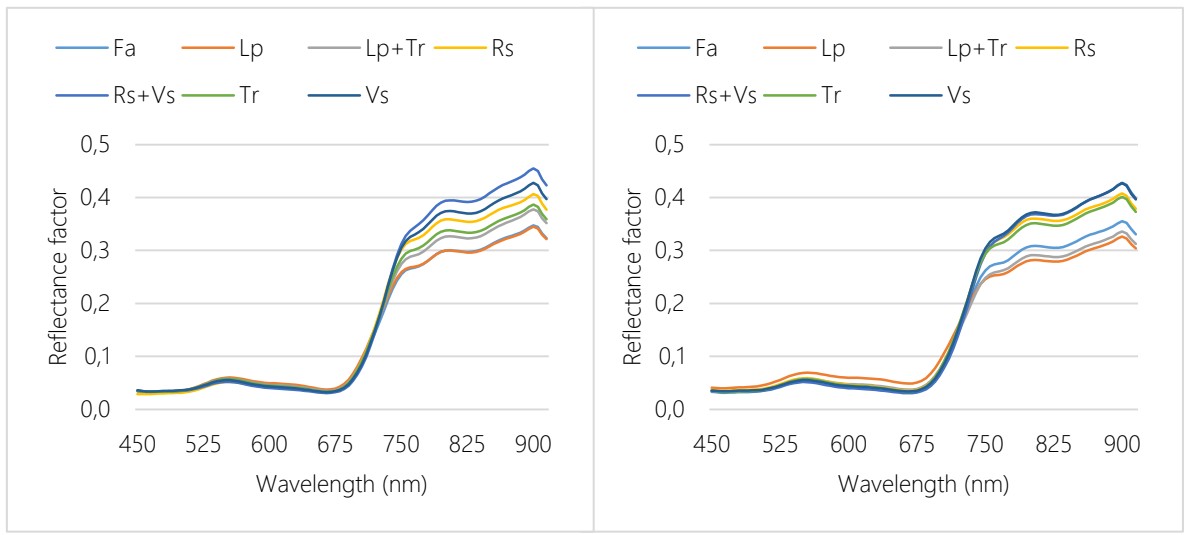

**Figure 4**: Average reflectance spectrum of oat grown in the different experimental plots and their associated treatments for the calibration (n=28, left) and validation set (n=28, right). Plant legacy treatments are: Fa = fallow, Lp = *Lolium perenne*, Rs = *Raphanus sativus*, Tr = *Trifolium repens*, Vs = *Vicia sativa*).

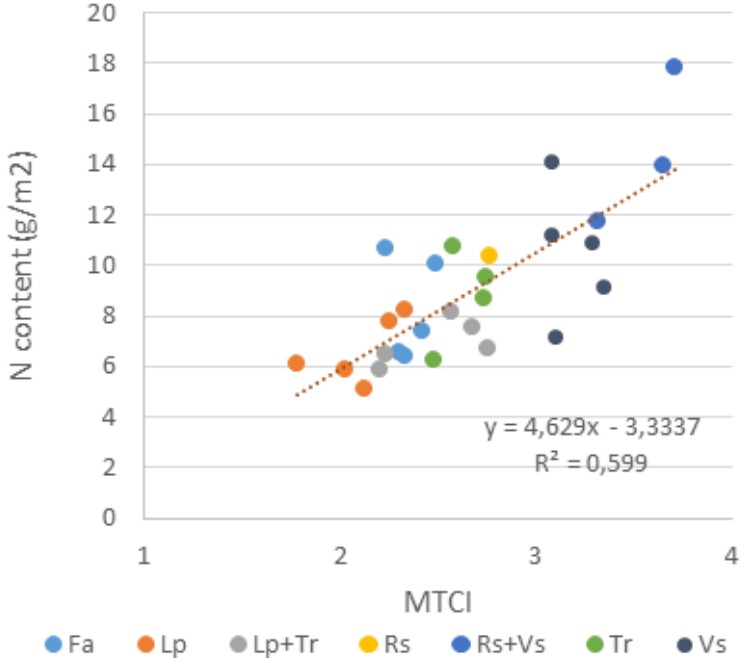

**Figure 5a.** Relation between field measured nitrogen content and index MTCI of *A. sativa* in response to plant legacy treatments: Fa = fallow, Lp = *Lolium perenne*, Rs = *Raphanus sativus*, Tr = *Trifolium repens*, Vs = *Vicia sativa*.

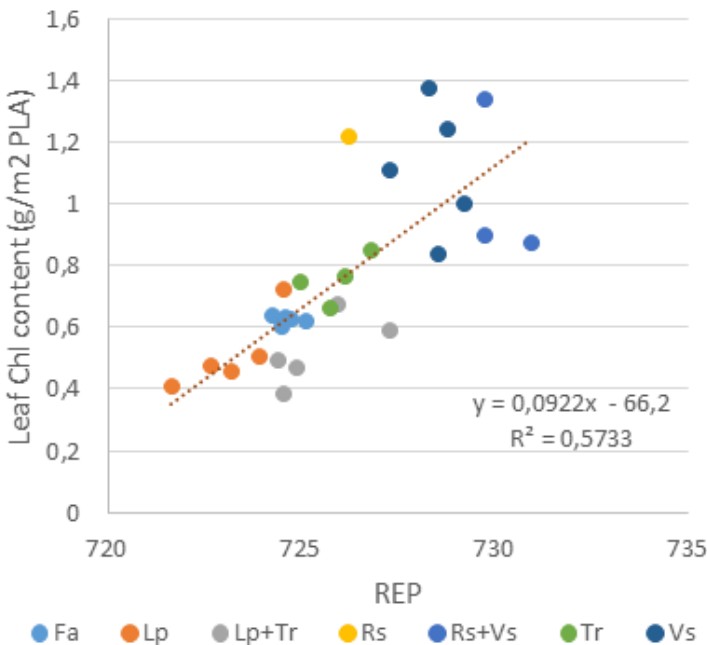

**Figure 5b.** Relation between field measured leaf chlorophyll content and Red Edge Position index of *A. sativa* in response to plant legacy treatments: Fa = fallow, Lp = *Lolium perenne*, Rs = *Raphanus sativus*, Tr = *Trifolium repens*, Vs = *Vicia sativa*.

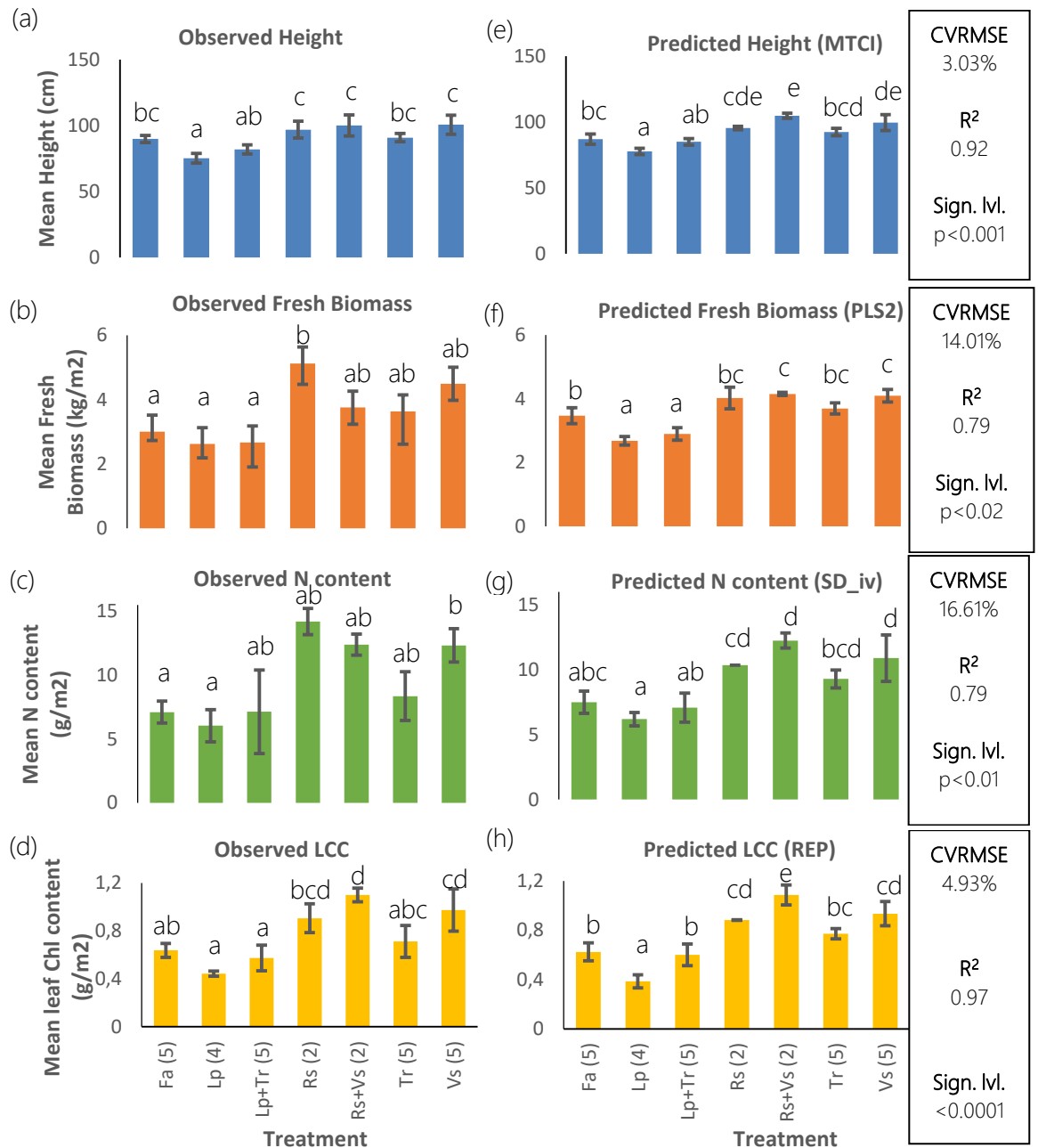

**Figure 6.** Mean and standard deviations of observed (a-d) and predicted (e-h) plant trait values in *A. sativa* per plant legacy treatment for the validation plots (n= 28). Bars with different letters above them indicate that the treatments are significantly different at p< 0.05 for the respective plant trait. The statistics to the right hand of the figure relate to statistical interference of the means of the observed and predicted values (Plant legacy treatments: Fa = fallow, Lp = *Lolium perenne*, Rs = *Raphanus sativus*, Tr = *Trifolium repens*, Vs = *Vicia sativa*, CVRMSE = Coefficient of Variation of the Root Mean Square Error, $R^2$ = Coefficient of Determination, Sign. lvl. = Significance level).