# Peer review of "Remote sensing of plant trait responses to field-based plantsoil feedback using UAV-based optical sensors"

_Biogeosciences, 2016_

## Referee Comment (RC1) · Anonymous Referee #1 · 12 Dec 2016

The paper is well written and logically structured. A plant-soil feedback experiment inducing variation in the growth of Avena sativa (oat) is used for the development of a model predicting plant height biomass, N-content and chlorophyll content. The models (for each plant characteristic) were build using hyperspectral and DSM information derived from a UAV flight. The model was built from a calibration dataset and validated on a validation dataset derived from the same population. Only one UAV flight was executed around the time of maturation of the plants as mentioned by the authors. The model building is well described as is the effect on predicted values and the subsequent statistical analysis (Fig. 6). In the title 'field-based plant soil feedback' is mentioned but no real biological interpretation related to the preceding crop is

given. This experiment was used to assess variability in plant characteristics as such I would omit this in the title and put 'Avena sativa' instead. The model building is well described. Further/Future improvements could be using a different flight, derived from a subsequent day, as a validation data set or use a bootstrapping method to find the best combinations of indexes or even use machine learning techniques based on the wavelengths. It is to be expected that there is a lot of redundancy to be found between the tested NDVI indices. Other combination of indices will perform as good or almost as good this could be discussed. Some minor issues are: - describe the RTK-GPS used: type, company, country - describe how plant height was measured e.g. from soil level to the tallest stretched leaf or... - p7l36: the sentence is unclear, probably a word is missing - p9l30: 'biophysical and biochemical oat plant constituents'. I would replace 'constituents' by 'characteristics' - p9l34: F-values are reported except for N content, why? - p9l35: the authors report that 'similar results' were found related to the F-values. If you compare the F-values, differences can be found resulting in a better post-hoc differentiation of the treatments. This is the case e.g. for fresh biomass: 4.93 vs. 24.58 or for Chl content: 11.10 versus 26.91. This should be more discussed. - p10l27: '2008; ' – '; ' can be removed - Fig. 3: a colour legend of plant height should be added

---

## Referee Comment (RC2) · M. Tuohy (Referee) · 19 Dec 2016

The authors have carried out a detailed study and presented a well written report on the outcome. Previous research has been thoroughly reviewed and the methods used have been well described. The conclusion that UAV-mounted hyperspectral sensors can adequately quantify plant traits may be a leap of faith considering that the best $R^2$ values for fresh biomass and N content were only 0.56 and 0.68 respectively. The PSF results could have been explained better; it is not clear what a good $F_{6,21}$ value is and the range varies from around 11 to almost 27. It could be argued that reflectance is not a good proxy for plant height and will never be, but it might well be expected to provide some measure of nutrient concentration. With the obvious importance of the

[Figure]

NIR wavelengths, perhaps more attention should be paid to this region of the spectrum rather than waste processing time on PLS analysis of all the bands.

Grammatical corrections. 3/32 replace good with well; delete remote based 4/12 of the field's 4/36 weighing not weighting 4/37 change to once in each plot. 5/1 ground not grinded; change to weighed in tin cups and then... 5/17 found to be inadequate 5/32 replace conflicting with conflict 6/31 replace was with were; change 'and using' to and a non-parametric... 9/6 replace till with to 11/23 use a more extensive... Colours in figs 5 and 6 should match those of the spectra in fig 4 Fig 6: small letters above each bar are not explained.

---

## Author Response (AR1)

The paper is well written and logically structured. A plant-soil feedback experiment inducing variation in the growth of Avena sativa (oat) is used for the development of a model predicting plant height biomass, N-content and chlorophyll content. The models (for each plant characteristic) were build using hyperspectral and DSM information derived from a UAV flight. The model was built from a calibration dataset and validated on a validation dataset derived from the same population. Only one UAV flight was executed around the time of maturation of the plants as mentioned by the authors. The model building is well described as is the effect on predicted values and the subsequent statistical analysis (Fig. 6).

Reply: We thank the referee for these positive comments on our manuscript.

In the title 'field-based plant soil feedback' is mentioned but no real biological interpretation related to the preceding crop is given. This experiment was used to assess variability in plant characteristics as such I would omit this in the title and put 'Avena sativa' instead.

Reply: In our study we tested the variability in plant traits of A. sativa in response to the legacies of the preceding crops. Therefore we prefer to retain the title and will provide a more extensive biological interpretation in the discussion part of our manuscript. The plant-soil feedbacks are generated via nutrient mineralisation/immobilisation which supports/constrains plant growth and these are linked to different organic matter inputs resulting from the cover crop treatments. Also the build-up of plant growth suppressing organisms can suppress plant height, biomass and nitrogen content, these effects however are more patchy/less homogeneous than plant-soil feedbacks generated via nutrient cycling.

The model building is well described. Future improvements could be using a different flight, derived from a subsequent day, as a validation data set or use a bootstrapping method to find the best combinations of indexes or even use machine learning techniques based on the wavelengths.

Reply: We agree that further improvements of our model building are possible and should be explored. We will provide these in the discussion by including the paragraph 4.4 Future improvements, these will include using data of several flights to improve temporal resolution, data analysis via bootstrapping and machine learning, and more accurately aligning field sample locations with UAV spectrometer data from which data is further processed, and improving the spatial sampling to also capture within plot variation. We will also include extra references:

- Capolupo, A. et al. (2015) Estimating Plant Traits of Grasslands from UAV-Acquired Hyperspectral Images: A Comparison of Statistical Approaches. ISPRS Int. J. Geo-Inf., 4: 2792-2820.
- Souza, A.A. et al. (2010) Relationships between Hyperion-derived vegetation indices, biophysical parameters, and elevation data in a Brazilian savannah environment. Remote Sensing Letters, 1: 55-64.
- Singh, A. et al. (2016) Machine Learning for High-Throughput Stress Phenotyping in Plants. Trends in Plant Science, 21: 110-124.

von Bueren, S. et al. (2014) Comparative validation of UAV based sensors for the use in vegetation monitoring. Biogeosciences Discussions, 11: 3837-3864.

It is to be expected that there is a lot of redundancy to be found between the tested NDVI indices. Other combination of indices will perform as good or almost as good this could be discussed.

Reply: We included the range of different NDVI indices because these have all been reported in literature and tested for one or two plant traits, whereas we wanted to explore how well these indices performed across a wider range of plant traits. We agree that redundancy between the indices can be expected but as it was not a priory clear which ones would produce the best results for our range of plant traits we decided to test the available indices as well as new combinations of two spectral bands in SR, SD and NDV indices. We will include in our discussion (section 4.2 Plant traits and physiological stage) that we tested a range of indices because the best fitting index was not a priory known and may differ depending on plant physiological stage.

Some minor issues are:

- describe the RTK-GPS used: type, company, country Reply: Included

- describe how plant height was measured e.g. from soil level to the tallest stretched leaf or. . . Reply: Indeed from soil level to the top of the plant, we include this now in material and methods.

- p7l36: the sentence is unclear, probably a word is missing

Reply: We rephrase the sentence into '...although indices yielding comparatively high coefficients of determination in relation to a distinct trait were generally found to also be rather strongly correlated to multiple of the other studied traits'.

- p9l30: 'biophysical and biochemical oat plant constituents'. I would replace 'constituents' by 'characteristics'

Reply: Replaced as suggested.

- p9134: F-values are reported except for N content, why?

Reply: The data of the in situ measured N content did not meet the assumptions for using parametric tests (variances were unequal also after data transformation). Hence we performed non-parametric tests which do not yield an F-value, we report the  $\chi^2$  value instead.

- p9l35: the authors report that 'similar results' were found related to the

F-values. If you compare the F-values, differences can be found resulting in a better

post-hoc differentiation of the treatments. This is the case e.g. for fresh biomass: 4.93

vs. 24.58 or for Chl content: 11.10 versus 26.91. This should be more discussed.

Reply: We do not explicitly compare F-values as such, we do compare whether or not the differences between the plant legacy treatments can be picked-up and whether the same treatment levels are being discriminated using the in situ measured data on the one hand and the remote sensed and modelled data on the other hand. We rephrased this part of the results to make clear what we mean: 'Similar results were found when using the predicted plant trait values from the remote sensing data to test the soil legacy effects: we found significant effects of plant legacies on oat plant height ( $F_{6,21}$ = 18.05, p< 0.001), fresh biomass ( $F_{6,21}$ = 24.58, p< 0.001), leaf chlorophyll content ( $F_{6,21}$ = 26.91, p< 0.001) and N content ( $F_{6,21}$ = 11.87, p< 0.001) (Fig. 6e-h).'

**-p10l27: '2008; ' – '; ' can be removed Reply: We removed the ;**

- Fig. 3: a colour legend of plant height should be added. Reply: We now included a colour legend for the figure showing plant height.

**M. Tuohy (Referee) Referee #2**

M.Tuohy@massey.ac.nz Received and published: 19 December 2016

The authors have carried out a detailed study and presented a well written report on the outcome. Previous research has been thoroughly reviewed and the methods used have been well described.

Reply: We thank the referee for these positive comments on our manuscript.

The conclusion that UAV-mounted hyperspectral sensors can adequately quantify plant traits may be a leap of faith considering that the best R2 values for fresh biomass and N content were only 0.56 and 0.68 respectively.

Reply: We thank the referee for the critical comment. However, we did not use the specific wording as suggested by the referee stating 'adequately quantify', we do state that the methodology offers great potential as we were able to discriminate between the treatments and obtained surface level information of a number of plant traits, in contrast to the point observation data of the *in situ* measurements which limit the spatial resolution.

The PSF results could have been explained better; it is not clear what a good F6,21 value is and the range varies from around 11 to almost 27.

Reply: The results of the PSF comprise the outcomes of the statistical tests in which we performed analysis of variance of the different plant traits in relation to the different treatments we imposed in the field by means of growing different species and species combinations of cover crops before growing oat. The significance of the F values is indicated

by the p values that are mentioned with it, with a p value < 0.05 indicating that the cover crop treatments resulted in different values of the plant trait of focus in the following oat crop. We expanded our description of the PSF effects in the results in order to clarify the findings.

It could be argued that reflectance is not a good proxy for plant height and will never be, but it might well be expected to provide some measure of nutrient concentration. With the obvious importance of the NIR wavelengths, perhaps more attention should be paid to this region of the spectrum rather than waste processing time on PLS analysis of all the bands.

Reply: The UAV based camera system used in this research includes both a hyperspectral and RGB sensor. The Structure-from-Motion method enables the derivation of a digital surface model (DSM) from the RGB images and from that to derive the plant height. The hyperspectral reflectance data were indeed used for deriving indices for plant chemical composition. As our work was in part explorative we included a range of PLS analyses, these however did not take up much processing time as we could run the analyses in a semi-automated way.

Grammatical corrections. 3/32 replace good with well; delete remote based Reply: Changed

4/12 of **the** field's Reply: Included 'the'

4/36 weighing not weighting Reply: Changed

4/37 change to once in each plot. Reply: Changed

5/1 ground not grinded; change to weighed in tin cups and then. . . Reply: Changed

5/17 found to be inadequate Reply: Changed

5/32 replace conflicting with conflict Reply: Changed

6/31 replace was with were; change 'and using' to and a non-parametric. . . Reply: Changed

9/6 replace till with to Reply: Changed

11/23 use a more extensive. . . Reply: We included 'a'.

Colours in figs 5 and 6 should match those of the spectra in fig 4 Reply: We adjusted the colour scheme for the different treatments in Figure 5 in order for it to match with the colour scheme of the treatments in figure 4. In figure 6 the colours we used relate to the different plant traits that we are addressing in the different panels, the colours hence do not relate to the different treatments as these are indicated in the x-axis of each panel.

Fig 6: small letters above each bar are not explained.

Reply: We had included the meaning of the small letters in the second sentence of our figure legend but the formulation may not have been clear enough. We therefore reworded this sentence into: '
[revised manuscript text omitted]

- *pronounced at the later waveenging.* This for later handwas predication of those that were constructed with 2*perenne*, *T. repens* or their combination (Lp+Tr) consistently exhibited the lowest canopy reflectance of *A. sativus* in the near-infrared. In contrast, near-infrared reflectance was highest for plots previously cultivated by *R. sativus*, *V. sativa* or a combination of these two cover crop species. *In situ* sampling also recorded the highest values for fresh biomass for these treatments (Fig. 4).
- 2.5 Values for fresh biomass for these treatments

**3.3 Univariate trait correlation with crop surface model (CSM) height**

CSM height was positively correlated to all crop traits, particularly for validation plots (Table 3). In general, the observed interdependencies confirmed the associated relationships between vegetation height and variables such as growth rate, biomass and plant fertility/health (e.g., Cornelissen et al., 2003; Tilly al. 2014). Strongest correlations were observed for *in situ* measured crop height, indicated by correlation coefficients of 0.85 and 0.91 for calibration and validation data, respectively. Furthermore, relative variations in CSM height were also significantly (p < 0.001) related to *in situ* measured height discrepancies for different treatments ( $R^2 \approx 0.95$ , NRMSE  $\approx 27.4$  %). The CSM, however, exhibited some bias and underestimated *in situ* measurements by 20cm on average. The other plant traits, i.e. N content ( $r \approx 0.69/0.73$ ), LCC ( $r \approx 0.67/0.79$ ) and fresh biomass ( $r \approx 0.57$ ).

35 0.62/0.74), displayed slightly lower correlation coefficients.

**3.4 Calibration**

10

**3.4.1 Relationship between existing vegetation indices (VIs) and crop traits**

In situ measurements were linearly regressed with a selection of well-established VIs (Table 1) based on the best matching bands from the HDC, the main product of hyperspectral mapping system. Regression analysis 5 yielded highly varying R2 values for different combinations of traits and existing VIs (Table 4), although indices yielding comparatively high coefficients of determination in relation to when correlated with a distinct trait were generally found to also be rather strongly correlated to multiple other traits. The relationship between in situ measured crop traits and VIs was strongest for the REP and MTCI indices, particularly for height ( $R^2 \approx 0.69$ ), N content ( $R^2 \approx 0.59$ ) (Fig. 5a), LCC ( $R^2 \approx 0.58$ ) (Fig. 5b) and, albeit to a lesser degree, for fresh biomass ( $R^2 \approx 0.58$ ) 0.25). The performance of these indices was closely followed by some of the evaluated two-band indices, NDVI in particular (Table 4). Exponential fitting improved R2 values for LCC in particular, albeit marginally, by 0.06 at most in some instances. This relatively minor improvement invoked by exponential fitting, compared to findings in previous studies, may be the result of the relatively limited range of LCC values.

In agreement with the wavelength dependency of REP (670 nm, 700 nm, 740 nm, 780 nm) and MTCI (680 nm, 15 710 nm, 755 nm), the best performing two-band indices recurrently exploit the near-infrared (> 750 nm) and the far red (± 710 nm), the red-edge (between 710 nm and 750 nm) and the far red, or solely the red-edge. Contrastingly, indices that performed relatively weak appeared to be primarily based on wavelengths in the visible part of the spectrum, particularly in the green (± 550 nm) and the blue. Soil background noise mitigating indices (i.e. TACRI/OSAVI and MACRI/OSAVI) did not enhance performance compared